# Health Benefits, Food Applications, and Sustainability of Microalgae-Derived N-3 PUFA

**DOI:** 10.3390/foods11131883

**Published:** 2022-06-25

**Authors:** Yanjun Liu, Xiang Ren, Chao Fan, Wenzhong Wu, Wei Zhang, Yanwen Wang

**Affiliations:** 1INNOBIO Corporation Limited, No. 49, DDA, Dalian 116600, China; liuyj@innobio.cn (Y.L.); fanc@innobio.cn (C.F.); wu@innobio.cn (W.W.); 2DeOxiTech Consulting, 30 Cloverfield Court, Dartmouth, NS B2W 0B3, Canada; wzhang14@gmail.com; 3Aquatic and Crop Resource Development Research Centre, National Research Council of Canada, 550 University Avenue, Charlottetown, PE C1A 4P3, Canada

**Keywords:** microalgae, n-3 PUFA, EPA, DHA, sustainability, health benefit, cardiovascular disease, neurodegenerative disease, food application, antioxidant

## Abstract

Today’s consumers are increasingly aware of the beneficial effects of n-3 PUFA in preventing, delaying, and intervening various diseases, such as coronary artery disease, hypertension, diabetes, inflammatory and autoimmune disorders, neurodegenerative diseases, depression, and many other ailments. The role of n-3 PUFA on aging and cognitive function is also one of the hot topics in basic research, product development, and clinical applications. For decades, n-3 PUFA, especially EPA and DHA, have been supplied by fish oil and seafood. With the continuous increase of global population, awareness about the health benefits of n-3 PUFA, and socioeconomic improvement worldwide, the supply chain is facing increasing challenges of insufficient production. In this regard, microalgae have been well considered as promising sources of n-3 PUFA oil to mitigate the supply shortages. The use of microalgae to produce n-3 PUFA-rich oils has been explored for over two decades and some species have already been used commercially to produce n-3 PUFA, in particular EPA- and/or DHA-rich oils. In addition to n-3 PUFA, microalgae biomass contains many other high value biomolecules, which can be used in food, dietary supplement, pharmaceutical ingredient, and feedstock. The present review covers the health benefits of n-3 PUFA, EPA, and DHA, with particular attention given to the various approaches attempted in the nutritional interventions using EPA and DHA alone or combined with other nutrients and bioactive compounds towards improved health conditions in people with mild cognitive impairment and Alzheimer’s disease. It also covers the applications of microalgae n-3 PUFA in food and dietary supplement sectors and the economic and environmental sustainability of using microalgae as a platform for n-3 PUFA-rich oil production.

## 1. Introduction

Microalgae as a food source have been consumed for thousands of years in different geographic locations and across different cultures to provide nutrients with good safety records [1,2]. However, it was not until the last few decades that the discovery of microalgae with high productivity of polyunsaturated fatty acids (PUFA) was reported, and soon after, scientific research and technological innovations enabled the microalgae-based PUFA products for commercialization [3].

Regarding PUFA, n-3 eicosapentaenoic (EPA) and docosahexaenoic (DHA) acids have been extensively studied and demonstrated for their health benefits to many ailments [4,5]. It is known that EPA and DHA can be produced endogenously from the precursor, α-linolenic acid (ALA) in humans [6], but the rate of biosynthesis is low and insufficient to meet the nutritional and physiological requirements [7]. Therefore, it is recommended to supplement EPA and DHA in diet or through a dietary supplement to provide beneficial effects on development, especially the neural and visual systems [4], and to mitigate a number of pathological conditions such as coronary artery disease, hypertension, diabetes, inflammatory and autoimmune disorders, neurodegenerative diseases, including Alzheimer’s disease, Parkinson’s disease, and depression [4]. However, oils rich in n-3 PUFA are in short supply as there is simply not a sufficient amount from the traditional sources to meet the fast expansion of the global market. Moreover, aquaculture as an alternative to the marine fisheries, although providing n-3 PUFA, exhausts a substantial portion of EPA- and DHA-rich oils and complete with humans and pets for n-3 PUFA oils, typically fish oil [8,9]. In the past decades, marine over-fishing catches have occurred. In addition, climate change has resulted in the decline of marine microalgae from which oily fish species obtain their n-3 PUFA, especially DHA and EPA [10]. Therefore, PUFA-rich microalgae have attracted considerable academic and industry interests concerning developing new supplies of n-3 PUFA as an alternative to fish oil [3,10,11,12].

The present review attempts to provide a broad insight on the health benefits of n-3 PUFA and cover the issues with the food/feed applications of microalgal n-3 PUFA and their sustainability. The delivery systems of n-3 PUFA-rich oils are discussed, together with the technologies used to protect n-3 PUFA from oxidation. The authors are strongly inclined to a holistic approach towards the solutions concerning non-drug interventions for Alzheimer’s disease (AD) or pre-AD, which include but are not limited to nutrients supplementation together with lifestyle management, social activity engagement, as well as biomarker monitoring and control strategies. The importance of formulation in nutrition intervention is especially emphasized. In addition, sustainability and the environmental impact of the n-3 PUFA industry is critical to resolve the conflict between the limited supply and the rapid increase in the n-3 PUFA-rich oil market. However, there is limited information currently available in these areas. Bearing this in mind, we are interested in filling up this gap, at least in part, by looking into the three inter-related aspects of the sustainability of microalgal n-3 PUFA production and application industries, including techno-economic, socio-economic, and environmental impacts.

## 2. Literature Search and Analysis

A comprehensive literature search was conducted by accessing several databases, such as PubMed, ISI-Medline, and Google Scholar. The keywords were chosen based on the key objectives of this review paper that are reflected in the title, subtitles, and contexts. Published papers in the past 20 years have primarily been selected, mirroring the health benefits, food applications, and sustainability of microalgae-derived n-3 PUFA. A total of 287 references have been included in this review. In each section, the selected references were summarized and analyzed to support our statements or help to explain the differences or discrepancies between different studies. We also attempted to draw our conclusions and point out the strengths and weaknesses, and accordingly the areas or directions that can be addressed by future research and technology/product developments.

## 3. The Health Benefits of n-3 PUFA

Growing evidence provided by numerous scientific and clinical studies demonstrates the health benefits of n-3 PUFA, particularly EPA and DHA. The major health benefits of these fatty acids are centered on the reduction of cardiovascular disease (CVD) risks, including high triglyceride level, high blood pressure, inflammation, as well as the improvement of cardiac and vascular functions [4,10,13,14,15,16,17]. In addition, n-3 PUFA change to play an essential role in promoting visual and brain function development of infants, children, and adolescents [18,19,20,21]. Since n-3 EPA and DHA are the precursors of anti-inflammatory mediators (such as eicosanoids and resolvins), an optimal or increased consumption of these fatty acids can help to alleviate the symptoms of neurodegenerative diseases and mental disorders, such as Alzheimer’s disease, Parkinson’s disease, and depression [22,23]. Other health benefits of n-3 PUFA include the protection of infection, prevention of autoimmune diseases, for example, rheumatoid arthritis, and various forms of cancers [23]. However, the human body cannot synthesize these essential fatty acids sufficiently because in vivo conversion rate from ALA is low. Accordingly, the intake of n-3 EPA and DHA through diets or supplements becomes important [16,24].

### 3.1. Cardiovascular Diseases

Early research dating back to the 1970s revealed a low rate of cardiovascular diseases in the Greenland Inuit and other ethnic groups, such as Japanese, who eat a fair amount of fish and other marine species [25,26,27]. Since then, numerous studies have been conducted, of which many generated positive results supporting the association between fish intake or dietary supplementation of n-3 PUFA and the incidence of cardiovascular events, such as heart attack and stroke [28]. It is quickly emerging that n-3 PUFA are able to reduce the cardiovascular risks by lowering blood pressure and triglyceride levels and inhibiting platelet aggregation, along with many other beneficial effects [29,30,31]. It is worth noting that the evidence of health benefits of n-3 PUFA is stronger for the secondary than the primary preventions [28].

In the early 2010s, several clinical trials in patients with CVD failed to replicate the previous positive findings and ignited debates over the health benefits of n-3 PUFA to CVD events [4]. For example, the 2010 Alpha Omega Trial used a low-dose supplementation of n-3 PUFA (226 mg EPA and 150 mg DHA a day) for 40 months and showed no reductions in the rate of major CVD events. In this study, the recruited 4837 patients all had previous cardiovascular events and were taking various medications [32]. Another example is the 2012 Outcome Reduction with an Initial Glargine Intervention trial in 12,536 patients who either had diabetes or were at a high risk for developing CVD [33]. After 6 years of supplementation of n-3 PUFA at a dose of 1 g/day (465 mg EPA and 375 mg DHA), the patients’ triglyceride levels were significantly lowered, but no effect were observed on the risk of myocardial infarction, stroke, or death from cardiovascular diseases [33]. The controversial results have been attributed to several reasons, such as small sample size, low dose of EPA + DHA, short study and follow-up periods, the low rate of CVD events that occurred during the study and follow up periods, previous CVD events in some patients, and the use of different medications [32,33,34].

Interesting results were obtained in two trials published in 2018, which include (1) the Vitamin D and Omega-3 Trial (VITAL) [35], and (2) the Study of Cardiovascular Events in Diabetes (ASCEND) [36] and one trial reported in 2019, which is the Reduction of Cardiovascular Events with Icosapent Ethyl-Intervention Trial (REDUCE-IT) [37]. A dose of 1 g/day of n-3 PUFA (460 mg EPA and 380 mg DHA) was used in both the VITAL and ASCEND studies. After a follow up of 7.4 years, the ASCEND study showed no effect on the risk of serious vascular events. A significant reduction (−19%) of cardiovascular death and protection against coronary events was observed in subjects who took 1 g/day of n-3 PUFA in comparison to the placebo controls [36]. Similar findings were observed in the VITAL trial, which showed no significant reductions in the rate of major cardiovascular events, but a significant reduction (−28%) of the total myocardial infarctions in subjects treated with n-3 PUFA in a study period of 5.3 years [35]. Strong effects of n-3 PUFA supplementation were noticed in subjects who consumed less than 1.5 servings of fish per week relative to those who ate more than 1.5 servings of fish in their day life. In this subgroup of the participants, a 40% reduction was realized by n-3 PUFA supplementation as compared with the placebo group [35]. It is suggested that the benefits of n-3 PUFA supplementation depend on the blood baseline of n-3 PUFA or quantity of fish edit by subjects in their regular life. The more fish they eat, the less benefits of n-3 PUFA supplementation are induced. It is also indicated that an optimal blood or tissue n-3 PUFA concentration is required in relation to individual health and disease conditions and could be personalized in order to optimize the benefits of n-3 supplementation [38].

The 2019 REDUCE-IT trial set up a milestone with the high dose (4 g/day) prescription drug Vascepa^®^ in which EPA was the sole source of n-3 PUFA, provided in an ethyl ester form [37]. Despite the high levels of triglyceride and LDL cholesterol levels in all 8179 patients, a 25% reduction in cardiovascular events was noted, which included cardiovascular death, nonfatal myocardial infarction, nonfatal stroke, coronary revascularization, and unstable angina. In addition, significant reductions were seen in other clinical outcomes, such as cardiovascular death by 20%, fatal or nonfatal stroke by 28%, and fatal or nonfatal myocardial infarction by 31% [37]. These results have attracted great attentions, leading to a reevaluation on the health benefits of EPA and DHA alone and their combinations [4,23].

In 2021, two meta-analyses provided new insights into the benefits of EPA and DHA on cardiovascular system, the effective dosages, and the effect of EPA alone verses the combination of EPA and DHA [39,40]. The meta-analysis conducted by Bernasconi et al. concluded that supplementation of EPA and DHA was an effective lifestyle strategy for CVD prevention and the efficacy appeared to be increasing with the dosage. This study covered many randomized control trials with EPA/DHA interventions and cardiovascular outcomes published before August 2019, with a total of 40 studies and 135,267 participants being included [40]. Khan et al. performed a meta-analysis of 38 randomized controlled trials on the cardio-protection of n-3 PUFA in 149,051 patients, stratified by EPA monotherapy and EPA + DHA therapy [39]. It was revealed that n-3 PUFA was associated with reductions of cardiovascular mortality, non-fatal myocardial infarction, coronary heart disease events, major adverse cardiovascular events, and revascularization. Another important finding of this meta-analysis was that EPA monotherapy showed a higher reduction than EPA and DHA combinational therapy in the random-effects rate ratios of all the aforementioned cardiovascular events [39]. A recent small randomized clinical study indicated that EPA and DHA work differently against chronic inflammation, with DHA being more potent than EPA; however, EPA is more efficient in balancing pro- and anti-inflammatory mediators [41]. The beneficial roles of EPA and DHA appear to be different and in some cases, one may block the beneficial actions of the other [4]. This finding has important clinical applications, and further studies are needed to verify the different roles of EPA and DHA in regulating inflammatory responses and cardiovascular health and events, and accordingly, the development of n-3 PUFA fortified food and beverages targeting cardiovascular health and disease prevention/treatment. Moreover, the health benefits of EPA and DHA seem to be dose-dependent, and promising results have been shown in some well-controlled clinical trials [42]. This information is highly valuable in modifying the recommendations while more studies are warranted.

Microalgae have advantages over fish as the sources of n-3 PUFA, in particular the production of singular EPA or DHA-rich oils. Microalgae are potentially excellent sources of singular EPA or DHA, avoiding the costly separation and purification processes involved in purifying or concentrating EPA or DHA from fish oil [3]. With the advances of species characterization, the application of bioengineering technologies in microalgae research, and the improvement of cultivation facilities and harvesting/processing technologies, a significant reduction in the production cost of EPA or DHA-rich oils using microalgae is expected.

### 3.2. Neurodegenerative Diseases

Numerous epidemiological studies have indicated that a long-term intake of the Mediterranean diet (emphasizing, amongst others, sea food, fruits, vegetables, and olive oil) is positively correlated with cognitive functions in aged populations [21,43,44,45]. In several longitudinal studies, increased intake of n-3 PUFA, measured by fish consumption, was associated with reduced risks for cognitive decline, AD, and dementia [46,47,48,49,50]. Therefore, dietary supplementation of n-3 PUFA has become a common practice of complementary and alternative medicine in preserving cognitive function. In 2010, the NIH Expert Consensus Opinions pointed out that risk reduction factors associated with neurodegenerative disease, for example AD, were limited from the nutritional and dietary aspects [51]. The available evidence does not support a clear association with most of the single nutritional or dietary components examined, except n-3 PUFA [51,52,53]. Several clinical trials were conducted to elucidate the roles of n-3 and other fatty acids in improving cognitive function and ultimately preventing or delaying the onset of AD [42,45,54]. Several systematic reviews and meta-analyses [55,56,57,58], including a Cochrane review [59], concluded that n-3 PUFA supplementation had no effects on cognitive function in healthy older adults or people with AD. However, for individuals with mild cognitive impairment, n-3 PUFA supplementation appeared to improve some aspects of cognitive function, such as attention, processing speed, and immediate recall [55,58,59].

Accumulative evidence suggests that the brain pathophysiological changes may begin 10 to 20 years before AD is diagnosed [60]. Once pathophysiological changes are initiated, it is difficult to stop, slow down, or reverse. With more than 100 potential mechanisms to be investigated, pharmaceutical companies have difficulties in developing effective ways to intervene the progression of illness [61], and consequently many therapeutics failed in their phase II or III clinical trials [62,63,64]. On the other hand, over the years, nutritional interventions have gained momentum in fighting against AD and other neurodegenerative diseases where n-3 PUFA have played significant roles [42,45,54,65].

It has become widely accepted that neurodegeneration leads to mild cognitive impairment and AD. This neurodegeneration process is very sophisticated and involves many potential mechanisms in which an effective cure with a single therapeutic agent has yet to be found [52,66]. This includes the monoclonal antibody Aducanumab approved most recently by FDA, which shows limited medicinal benefits in AD patients [52]. Similarly, from a nutritional perspective, it has been difficult to find a single nutrient that is able to prevent or delay the neurodegenerative process. Instead, it has been shown that consumption of the Mediterranean diet is inversely associated with CVD and has protective effects against hypertension, obesity, and AD [45]. A recent thorough systematic review and a meta-analysis suggest that the Mediterranean diet can reduce the risk for developing mild cognitive impairment and the progression from mild cognitive impairment to AD [67,68]. The results of several studies on the Mediterranean diet have suggested a synergistic effect of multiple nutrients on cognitive function through distinct mechanisms associated with a single or few of nutrients [52,69]. The Mediterranean diet is a complex dietary system consisting of multiple nutrients derived from fish, olive oil, nuts, and other food ingredients such as fruits and vegetables, with some of them not easy to be adopted by peoples in other cultures and geographic locations. A commercially available formula of multi-nutrients could be served as one of the options for those who want to take it as a dietary intervention to reduce the risk for developing mild cognitive impairment (MCI) and AD [45]. Souvenaid^®^ is a typical example of such formulae. It is also called Fortasyn Connect^®^) and is a formulated dairy nutritional beverage, developed commercially as a nutritional intervention against AD and dementia based on the research work conducted by Wurtman in 2006 [70]. Souvenaid^®^ was designed to support synapse formation and function in early AD [54]. DHA and EPA are the two major nutrients in Souvenaid^®^, and other nutrients include uridine monophosphate, choline, vitamins (B12, B6, C, E, and folic acid), phospholipids, and selenium. Several clinical trials with Souvenaid^®^ have shown inconsistent results, with insignificant effects being seen in patients with mild to moderate AD, but certain improvements noted in subjects with mild AD dementia or MCI due to AD pathology [49,71].

The Europe LipiDiDiet trial was designed to investigate the effects of Fortasyn Connect on cognition and related measures in prodromal AD in patients with MCI due to AD pathology [71]. The experiment was planned for 24 months originally, but encouraging data were obtained only when the trial was extended to 36 months and beyond. The Fortasyn Connect did not show a significant effect on the primary endpoints after 2 years of intervention, possibly due to the lower-than-expected cognitive decline in all the participants. However, significant improvements were observed in the outcomes of the secondary endpoints on several cognitive tests, which led to the extension of the trial to 36 months and beyond. After 36 months, significant reductions were observed in 6 clinical tests related to cognitive function, brain atrophy, and disease progression in the patients treated with the Fortasyn Connect [54]. The prolonged intervention from 24 to 36 months resulted in a broader range of endpoints that were different from that measured after 24 months. The lasting benefits of a three-year intervention with LipiDiDiet have not been reported previously in patients with MCI due to AD pathology, suggesting that early and long-term intervention might further increase the observed cognitive benefits [54].

In addition to multi-nutrients intervention, researchers have investigated a multi-domain approach for the prevention and treatment of AD and other neurodegenerative diseases [65]. A Multi-domain Alzheimer Preventive Trial (MAPT) was conducted to determine the effect of the combination of nutrition, physical exercise, and cognitive stimulation, together with n-3 PUFA supplementation on the cognitive functions, and the results showed that this approach was effective in slowing cognitive decline in frail older adults [72,73]. Further analysis revealed that participants with amyloid-β positive responded to the combined treatment but not n-3 PUFA alone [72,74]. The multi-domain intervention is promising as a long-term strategy in preventing the development of impaired cognition [72,73].

Many inventions have combined n-3 PUFA with other nutrients by targeting an array of brain health benefits in recent years. Fortier et al. [75] reported that brain energy rescue with medium chain triglycerides (MCT) is an emerging potential strategy to reduce cognitive decline in patients with MCI and AD. Researchers at the University of Sherbrook [76] recently conducted a randomized controlled Brain Energy, Functional Imaging, and Cognition (BENEFIC) trial. In this study, the patients with MCI given a ketogenic drink containing MCT for 6 months had improved cognitive outcomes. Hageman et al. [77] patented a lipid formula for brain function support, consisting of EPA, α-linolenic acid, and about 10% of MCT in total lipids. In another study, n-3 PUFA was combined with lycopene, carnosic acid, or lutein for their synergistic effects to inhibit/suppress neuroinflammation [78]. Power et al. [79] conducted a 12-month clinical trial to determine the potential synergistic effects of a combination of n-3 PUFA, carotenoids, and vitamin E on cognitive function in patients with MCI. The results showed significant improvements in episodic memory and global cognition in the treatment group compared to the placebo. Miquel et al. [42] summarized the research work around nutrition interventions for their effects on the AD types of neurodegenerative conditions and concluded that single-nutrient approaches (including n-3 PUFA, EPA, and DHA) are not able to show significant effects and a longitudinal approach is recommended. Furthermore, interventions may need to be imposed before the initiation of pathophysiological conditions. Other factors, such as epigenetic changes or gut microbiome diversity in the aging process, should be considered. A summary of some clinical trials for the effects of n-3 PUFA on neural development and neurodegenerative diseases is presented in Table 1.

### 3.3. Neurodevelopment in Infants and Children

It is known that DHA is highly concentrated in the human brain and retina, while only little EPA can be found in these tissues [85]. Therefore, an adequate supply of DHA is important for optimal neural and visual development in infant. In the 1980s and 1990s, studies in pre-term infants demonstrated the benefits of DHA and ARA fortified infant formulas on both visual and neural developments [86,87]. However, a recent meta-analysis of fifteen trials on DHA fortified infant formulas for term infants showed little evidence to support its benefits to visual and neural developments, even though a few studies seemed to show favorable results [88,89]. One of reasons for this discrepancy might be that the time effect or the effect was shown in the early examinations but was lost in the later tests. Regardless, it has been a global practice to fortify infant formulas with n-3 PUFA, especially DHA [90].

Several randomized controlled trials in adolescents with attention deficit hyperactivity disorder have shown improvements after n-3 PUFA supplementation [91,92,93]. However, clinical studies in children with attention, learning, or behavioral disorders and autistic spectrum disorder showed inconsistent results, and lower levels of EPA and DHA were found in the blood stream of those children [94,95,96,97,98]. These findings may suggest that supplementation of n-3 PUFA in children depends on their nutritional status or the blood concentration of n-3 PUFA. It is more likely to see the beneficial effects of n-3 PUFA supplementation in children who are deficient in n-3 PUFA or have lower levels than the recommended or so-called “healthy levels” defined by nutritionists and medical professionals based on the collection of relevant scientific evidence.

### 3.4. Depression

Depression is another area where n-3 PUFA has shown beneficial effects [99]. One study conducted in nine countries showed a significant correlation between dietary fish consumption and the occurrence of major depressive disorder [100]. A dose-dependent relationship has been found in some clinical studies where a high dose of EPA + DHA (9.6 g/day) resulted in the improvement of depression, while a lower dose of DHA (2 g/day) showed no effect [101,102]. Several meta-analyses have consistently shown the benefits of n-3 PUFA on major depressive disorder [103,104,105,106,107]. One interesting finding is the positive correlation between EPA intake and study outcomes, or a higher dose of EPA supplementation being linked to a better improvement of the depression symptoms [103,104]. A 2015 Cochrane review of 26 studies concluded that there was not enough evidence to support the beneficial effects of n-3 PUFA on the major depression in adults [108]. It is evident that not all studies have shown a consistent association between dietary fish consumption or n-3 PUFA supplementation and the major depression, while the majority of studies and clinical trials have indeed demonstrated the benefits of increased n-3 PUFA intake from fish consumption or dietary supplements.

### 3.5. Cancer Prevention and Treatment

The effect of n-3 PUFA on cancer prevention and treatment is complicated, with some studies showing a decreased risk, while others demonstrate an increased risk [90,109,110,111,112]. Inhibition of the inflammatory responses by n-3 PUFA has been linked to the reduction of cancer risk [113]. Several observational studies showed reductions in risk for developing breast cancer in subjects with higher intakes of n-3 PUFA [114,115]. Several meta-analyses on clinical studies concluded that the risk of breast cancer was negatively and dose-dependently correlated with the dietary intake of n-3 PUFA [116,117]. Interestingly, there was not an association between the intake of ALA or fish and cancer risk [116]. Although unclear, it is well-known that the conversion rate from ALA to n-3 PUFA, EPA, and DHA is low in the body. The content of n-3 PUFA in fish is largely varied, depending on fish species and geographical locations (or weather conditions) where fish are grown. Thus, the effect of fish consumption on cancer risk is dependent of n-3 PUFA content in fish meat or lipids.

Regarding colorectal cancer, several reviews and meta-analyses have concluded a negative association between fish or n-3 PUFA intake and colorectal risk [118,119,120,121]. When colon cancer and rectal cancer were distinguished, the reduction of risk was more substantial for rectal cancer than colon cancer [118].

The effects of n-3 PUFA on prostate cancer are controversial between studies, although a negative association was seen between n-3 PUFA intake and prostate cancer risk in some studies [112,122,123]. In the Prostate Cancer Prevention Trial, blood levels of DHA were positively linked with risk of high grade but not low grade types of prostate cancer, while no association was seen between EPA intake and the risk for either of the two grades of prostate cancer [122]. Other studies showed the opposite results where EPA was positively linked to the risk of prostate cancer, while DHA had no effect [124]. The different responses of the ethnic groups were also noticed in the Multi-ethnic Cohort Study where higher blood levels of n-3 PUFA were associated with an increased risk of prostate cancer in white participants, whereas no such relationship was noticed in the participants of other ethnic groups [124].

In a large clinical trial, no association was found between n-3 PUFA intake and reduction of the overall risk of cancers, including breast cancer, prostate cancer, or colorectal cancer [4,23,125,126,127,128,129]. However, dietary intake survey data suggest that fish or PUFA intake is linked to the reduction of cancer risks [130,131]. Further studies with well-designed clinical trials are required to demonstrate the effects of n-3 PUFA on the risk for the various types of cancer.

In general, n-3 PUFA has been of great research interest since the late 1980s. To date, numerous studies have been conducted to determine the beneficial effects of n-3 PUFA on nutritional homeostasis, metabolic and physiological functions, health conditions, and diseases [4]. Research aimed at better defining the roles of n-3 PUFA in human growth, development, function, health, well-being, disease risk, and better understanding of the underlying molecular and cellular mechanisms will continue in the foreseeable future. As can be seen in the previous context, there remain considerable inconsistencies and gaps in the literature that may not allow for robust recommendations and dosages of n-3 PUFA for healthy people and specific subjects or patients, although a daily intake of around 500-600 mg of n-3 PUFA has long been recommended. Several reasons might be responsible for the observed inconsistencies, including variations in study participants or amongst particular subgroups, sex, body composition, genetics, life stage, physiological state, health/disease status, medications, and the baseline of n-3 PUFA. Data on the different roles of n-3 PUFA and individual EPA, DHA, and other n-3 fatty acids are lacking [132,133]. In addition, as the effects of EPA and DHA are dose- and availability-dependent [134,135], it is necessary to develop more effective delivery systems, including but not limited to different chemical formulae [136,137] and physical [138,139] forms, and further on, the optimal doses targeting different population groups.

It should be recognized that n-3 docosapentaenoic acid (n-3 DPA) and n-6 DPA are also important PUFA but less studied, especially the latter. N-3 DPA is the metabolic intermediate of EPA and DHA and evidence suggests that n-3 DPA is an important contributor to total n-3 PUFA intake and imparts unique benefits [140]. Several studies have shown that n-3 DPA is a source of EPA and slightly DHA in the major metabolic organs, and also the precursor of a large panel of lipid mediators principally implicated in the pro-resolution of the inflammation with specific effects similar to that of the other n-3 PUFA [141]. N-3 DPA is the second abundant n-3 PUFA in the brain after DHA and can be specifically beneficial to elderly neuroprotection and early-life neuronal development [141]. N-6 DPA shares some of the physiological functions and benefits with n-3 DPA, while also plays different roles [142]. It was reported when the level of DHA fell in the brain and retina, there was a concomitant increase in the n-6 DPA [143,144], which was possibly a result of filling up the roles, at least in part, that 22-carbon n-3 PUFA essentially play. It is also demonstrated that LA metabolism is not sufficient to supply an adequate amount of n-6 DPA when n-3 fatty acids are deficient during the fast brain growth that occurs in the post-natal period [145]. The functional abnormalities that are associated with the n-3 PUFA deficient-diet may be related to the deficit of 22-carbon fatty acids; however, the abnormalities cannot be completely offset by n-6 DPA, even though n-6 DPA is supplied sufficiently or in excess. In a study which rats were fed n-6 LA or n-6 DPA, brain DHA was dramatically decreased (>60%), and the loss of DHA was largely compensated by n-6 DPA. Nevertheless, these rats exhibited a defect in spatial retention, although they had a longer escape latency than those given DHA [146]. Recently, n-6 DPA has been produced using microalgae and commercialized; however, the metabolic and physiological functions and impacts on health of this fatty acid has not been well studied [147,148,149], and therefore, further investigations are warranted to explore the potential benefits vs. negative effects of this fatty acid on human health.

## 4. Food Applications of Microalgae-Derived n-3 PUFA

As mentioned above, n-3 PUFA play crucial metabolic and physiological roles in different stages of life with significant health benefits and are indispensable to human nutrition and health [38,150]. The success of pharmaceutical grade EPA and DHA in treating hypertriglyceridemia [4], a high and potentially risky level of triacylglycerols, has been one of the driving interests in consumers for increasing n-3 PUFA intake [3], and more are on the numerous beneficial effects on the body systems, cellular functions, and various metabolic and physiological processes [151]. WHO and other health organizations in various countries/regions recommend a dietary intake of EPA and DHA at a range of 250 to 1000 mg/day [113] and the most commonly recommended dose is 500 mg/day [9]. Fish, as the primary source of EPA and DHA, cannot meet the global demand, resulting in a gap of nearly 1.1 million tons annually [152]. A further exploitation of natural fish and other marine sources for EPA and DHA is linked to the damage of ecosystems. Therefore, the traditional n-3 PUFA supplied from the fishery industry has encountered sustainability issues over the years [9]. Strategically, microalgae as a biological factory of n-3 PUFA have increasingly been used commercially, starting in the early 21st century, with several species and strains showing high productivity (Table 2). There is a wide range of the applications of microalgae as the sources of n-3 PUFA, such as infant formulae, nutraceuticals, dietary supplements, pharmaceuticals, foods, and beverages [3,153,154,155,156,157]. Numerous efforts have been invested, aiming at boosting productivity and improving processing technology, so as to produce a sufficient quantity of n-3 PUFA, which is also economically acceptable, to meet the fast expanding n-3 PUFA market worldwide [8,9] (Figure 1).

Neither human nor fish can synthesize EPA and DHA efficiently from their precursor ALA [10]. Marine algae and phytoplankton are the primary sources of n-3 PUFA, and marine fish obtain and accumulate EPA and DHA by eating them [163]. In aquaculture, fish, such as Atlantic salmon, obtain n-3 PUFA from their diets, and nowadays, microalgae have been increasingly studied for their use as a protein and n-3 PUFA source of fish diet. Intake from diet, together with de novo synthesis or conversion from the precursor, enable fish to accumulate high contents of EPA and DHA in flesh lipids. Microalgae can now be used to produce, at large scales, biomass or oil that contain a high content of n-3 PUFA. Technology advances enable the enrichment of n-3 PUFA in foods and beverages via different approaches, such as (1) addition of n-3 PUFA-rich oil purified from microalgae to food alone or combined with some food additives; (2) addition of n-3 PUFA-rich microalgal biomass to food; (3) dietary supplementation of n-3 PUFA-rich oil or n-3 PUFA-rich microalgal biomass, and (4) enrichment of animal products with n-3 PUFA by feeding the animals with n-3 PUFA-rich diets [164].

### 4.1. Addition of n-3 PUFA Oil to Food

Although the discovery of a valuable n-3 PUFA producing species dates back to the 1980s, it was not until the early 2000s when DHA-rich (up to 60% of total fatty acids) oil from microalgae *Crythecodinium cohnii* was realized commercially [3,8,165,166]. The product has primarily been used to fortify infant formulae [166]. Other DHA-producing microalgae species include *Schizochytrium* sp. and *Ulkenia* sp., of which *Schizochytrium* sp. is known for its ability of fast lipid production. This microalgae can increase the cell density to over 200 g/L within 72 h, with the lipid content over 40% of the biomass [167]. *Schizochytrium* sp. can produce a high yield of n-6 DPA (up to 17%), with n-3 EPA up to 19% of the total lipids [168]. Depending on lipid composition, some n-3 PUFA-rich oils derived from microalgae are used in infant formulae, while others are designated for other food applications, excluding baby foods [169]. Infant formula is the first type of commercially successful foods that are enriched with microalgae-derived n-3 PUFA, still taking a lion’s share of the microalgae n-3 PUFA oil market [12].

It should be realized that DHA can be retro-converted to EPA by enzymes in the human body, which offsets the health benefits of DHA in the brain development of infants, whereas ARA can suppress this conversion [170]. As such, it becomes a standard practice to mix microbial ARA-rich oil with microalgal DHA-rich oil, for example, at a ratio of 2:1 in the infant formulae [3]. Both oils were given a GRAS status by US-FDA in 2001 [12]. By 2010, Marteck’s *life’s*
*DHA*^™^ oils were sold to 24 infant formula producers and occupied more than 70% of the world infant formula market [171]. According to Global Market Insights Inc., the DHA algal oil for infant formula market was estimated at $245 million in 2019 and is slated to surpass $435 million by the end of 2026 (https://www.gminsights.com/request-sample/detail/4832; accessed on 17 March 2022).

Apart from infant formulae, microalgal n-3 PUFA-rich oils are used to enrich or fortify dairy products, such as liquid milk and yogurt [172], meat products [173], ground turkey patties, fresh pork sausage, and restricted ham [174,175]. In these products, n-3 PUFA-rich oils are not directly added but instead used in a form of oil-in-water emulsions, which provide a better stability against oxidation during food processing and storage, even at a low pH environment [176].

N-3 PUFA are highly unstable due to the presence of multiple double bonds in the backbone of the molecules. Oxidative degradation can result in poor product quality, such as off-flavors and odors [177,178], due to the generation of free radicals and oxidative products [179,180]. Transition metals naturally present in food can act as a catalyst to lipid oxidation [181]. In addition to various types of antioxidants that have been employed to enhance oxidative stability of n-3 PUFA in food applications [182,183,184], a number of microencapsulation technologies have been applied as well [165,185]. It has been proven that encapsulation of n-3 PUFA oil formulated with the addition of antioxidants is an effective strategy to enhance product quality [180,186]. In this regard, tocopherols combined with ascorbic acid or lecithin, rosemary extract, quercetin, chlorogenic acid, caffeic acid, and catechin are commonly used [180,187,188]. Microencapsulation technologies are relatively new and now widely used to encase n-3 PUFA-rich oils in layers of shell materials or embedded in a structured matrix, providing protections against fatty acid oxidation during food processing and storage [168]. It has been showed that some encapsulating wall materials possess antioxidant properties. One such example is whey protein hydrolysates that have antioxidant activities, such as radical scavenging activity, metal chelating, and reducing power [189]. Spray drying is one of the most frequently used microencapsulation technologies [177,190]. A recent study revealed that spray dried microcapsules, composed of 13% n-3 PUFA-rich oil stabilized by whey protein hydrolysates and glucose syrup, significantly enhanced the oxidative stability of n-3 PUFA during storage when used in low-fat mayonnaise [191].

### 4.2. Enahancement of n-3 PUFA Bioavaialbity

While n-3 PUFA can be enriched by adding n-3 PUFA-rich oil to food, the bioavailability of n-3 PUFA is critical in achieving the expected nutritional and physiological functions and the associated health benefits. In general, the delivery form determines the bioavailability of a nutrient or bioactive compound. For example, a randomized controlled trial was conducted for a period of 28 days to assess the bioavailability of DHA in microalgal DHA-rich oils from two different algal species in a form of capsules in comparison to a microalgal DHA-fortified food [192]. It was found that snack bars fortified with microalgal DHA oil delivered an equivalent amount of DHA compared to the capsule form. The same group compared the bioavailability of DHA in cooked salmon and algal-oil capsules, and the results demonstrated that algal DHA-rich oil in capsules and cooked salmon appeared to be bioequivalent in providing DHA in plasma and red blood cells [193]. Other studies showed the improved bioavailability and bioequivalence of microencapsulated EPA/DHA in different product forms, such as emulsion, powder, and fortified in food in comparison to softgel encapsulates or bulk oil [194,195,196]. One of the most recent innovations in the commercial delivery of n-3 PUFA-rich oils has been developed based on a self-microemulsifying delivery system, which is reported to improve n-3 PUFA bioavailability by several folds under fasting conditions compared with the regular forms, such as free fatty acid mixture or fatty acid ethyl ester concentrate in a capsule form [197,198]. It is apparent that microencapsulation is an effective technology for the improvement of n-3 PUFA bioavailability [199]. With advances in creating new knowledge and elucidating the parameters in formulation and further processing, novel microencapsulation techniques and technologies are expected to be created, which will further improve the n-3 PUFA bioavailability and provide insightful guidelines to expand the application of n-3 PUFA in foods and nutritional and health products [194,200].

### 4.3. Inclusion of n-3 PUFA-Rich Microalgae Biomass in Food

Human consumption of microalgae can be dated back to thousands of years ago when some species were demonstrated to be safe as human food [1,2]. *Nostoc*, Arthrospira (usually denoted as *Spirulina* in the market), and *Aphanizomenon* are protein-rich microalgae and have long been used to supplement protein in the human diet [160]. In the 16th century, Spanish chroniclers observed that Mexican Aztecs harvested microalgae from the surface of alkaline, dried it and consumed it directly as food without any health problems or symptoms of illness [201,202]. Aztecs started to cultivate and consume the cyanobacterium Spirulina (*Arthrospiraplatensis, Arthrospira maxima*) from Texcoco salted lakes around AD 1300 [203,204]. People in Chad have been harvesting Spirulina (*Arthrospira fusiformis*) from Lake Kossorom at the northeast fringe of Lake Chad and using it for food on a daily basis. This activity is believed to date back to perhaps the 9th century [205]. The harvested microalgae were sun dried and prepared as thin green wafers for direct consumption or to be powdered as an ingredient for a sauce, some of which are flavored, to be served on galls of cooked millet or maize [206,207]. *Nostoc flagelliforme, N. muscorum,* and *Nostoc sphaeroides* have been used by the Chinese as a food delicacy and for their herbal values for hundreds of years [208]. These species are also traditionally consumed by people in Mongolia, Tartaria, and South America [204]. *Aphanotheca sacrum* is found in Japan and during the Edo period, the domain of Hosokawa and Akizuki conserved *Aphanotheca sacrum* as a material of rare country dishes [209], and the biomass of this microalgae has been used in local cuisine in Kyushu, Japan, for over 250 years [210]. The filamentous green algae *Spirogyra* and *Oedogonium* are used as a dietary component in Burma, Thailand, Vietnam, and India [204]. The earlier consumption of microalgae either directly or via incorporation into other food metrics was mainly to provide nutrients to people who were malnourished or in poverty, primarily to provide proteins while providing other nutrients as well, such as lipids and n-3 PUFA [211]. However, health benefits were not recognized until later, when research demonstrated their cellular and metabolic functions.

Consumption of microalgae has been steadily rising. Adding microalgae to food products is a great strategy to enrich diet with n-3 PUFA and many other nutrients and beneficial components. Numerous examples exist in the incorporation of microalgae in food, which are demonstrated to be safe and effective in providing n-3 PUFA and other nutrients including protein and antioxidants [212]. Amongst them, cookies and biscuits have been considered as a suitable food matrix for incorporating n-3 PUFA-rich microalgae due to their good taste acceptance, versatility, and convenient consumption [213]. Pasta is another food delivery vehicle for n-3 PUFA-rich microalgae [214,215]. In addition, microalgae have been used to enrich vegetarian gels with n-3 PUFA and have shown antioxidant properties as well [216,217].

Although microalgae added to food have numerous health benefits, it is important to understand the digestion, absorption, and distribution of algal nutrients. Wells et al. (2017) examined the release mode of nutrients in algal food, their transports in digestive epithelium, the influence of the gut microbiome, transformations during the digestion process, and the final fraction that reaches the body system in a strain-specific manner. These effects can be manifested via translocation across the small intestinal epithelial cells into the blood stream, interactions with the digestive epithelia, altering uptake of other substances, regulating the microbial consortium, or contacting with colonic epithelial cells of the large intestine [2]. It is complicated, and more studies are needed to elucidate the interactions of microalgal components with the body systems and how they affect the bioavailability and bioefficiency of algal n-3 PUFA when added as a biomass in food. An emphasis should be put on the cell wall structure and composition, which are unique as compared with other plant species, and have been shown to significantly affect the release and digestion of intracellular nutrients [218,219], as cell wall disruption improves bioavailability, such as marine microalgae *Nannochloropsis gaditana* [220]. If a biomass is instead extracted or concentrated oil is used, pretreatment of biomass such as cell wall disruption is important to the extraction efficiency.

### 4.4. Dietary Supplement of n-3 PUFA-Rich Biomass or Oil

Dietary supplements have been gaining notable momentum in recent years for many reasons, and the main driving force is the increased concerns about chronic diseases and their associations with diet habits. Consequently, consumers are increasingly taking various supplements to promote their health and reduce the risk for developing diseases, aiming to improve life quality and lifespan [221,222]. By adding in foods and beverages, microalgae biomass is used nowadays predominantly in the health food market and reported to repsent 75% of the annual microalgal production [223]. Today, the most popular way to consume microalgae in adults is as a dietary supplement in capsule, tablet, or powder [224]. There are several companies performing nutraceutical research and product development based on microalgae oils that are rich in EPA and DHA, such as Algae, Parry Nutreaceuticals, Fermentalg, Algaetech International, AlgaeBiotech, Algae to Omega Holdings, Alltech Algae, etc. [222]. PureOne^TM^ is one of the algal oil supplements in the market which is rich in EPA and DHA and sold as the capsules. Source oil Algae Omega 3, extracted from *Schizochytrium* sp., is another n-3 PUFA supplement approved by the FDA for human consumption [221]. Nordic Naturals Algae Omega is rich in n-3 PUFA, prepaed from marine microalgae and packed in softgels, and is claimed to be suitable for vegetarians [221]. Algarithm produces a variety of n-3 PUFA oils that are packed in different forms with different n-3 PUFA profiles, such as Alphamega algal oil (40–53% DHA), Betamega algal oil powder (12% DHA), Kappamega algal oil softgels (400 and 530 mg algal DHA formulae), and Gammamega (custom n-3 PUFA emulsions) (www.algarithm.ca; accessed on 15 March 2022).

In addition to capsules and softgels, several studies were conducted to develop powder forms suitable for specific applications in tablet products of n-3 PUFA oils [225,226]. Currently, the commercially available n-3 PUFA powders are typically prepared by microencapsulating oil in biopolymers (often proteins and/or polysaccharides of food origins, [227,228,229]. It is reported that high pressure used to make tablets may cause the individual particles with the encapsulated oil droplets in biopolymers to burst and the broken particles can leak the oil out, resulting in unappealing appearance, unpleasant smell and taste, in addition to significantly reduced shelf life of the tablets made from these powders. Thus, further studies are carried out to develop new methods to make n-3 PUFA oil powders with improved crushing strengths and compatibility. A dry and compactible powder could be prepared from various n-3 PUFA oils and β-cyclodextrin, and spray granulation appeared to be the superior drying method for the preparation of compactible powders [226]. Evidently, developing new technologies and product forms is imperative in improving the compactible function of powder in order to make good quality tablets for the delivery of n-3 PUFA powder as a dietary supplement.

### 4.5. Feeding Farmed Animals with n-3 PUFA-Rich Biomass or Oil

Microalgae have been used for many years as a feed ingredient in fish and animals [11]. The potential benefits of adding n-3 PUFA-rich microalgae in aqua- and animal-feeds include the enhanced immune function, increased growth rate, and elevated n-3 PUFA contents in meat and other edible products [230,231]. In general, fish is superior to any terrestrial animals for providing n-3 PUFA-rich oil. However, fisheries supply is limited due to various reasons, and thus fish farming and aquaculture of other seafood species remain as the main food chains for the delivery of microalgal n-3 PUFA in humans [232]. As a matter of fact, several microalgal species have been assessed as the sources of n-3 PUFA, such as *Dunaliellatertiolecta* for scallop larvae [233], *Lobosphaeraincisa* for Zebrafish [234], *Nannochloropsis* sp. for Kuruma shrimp [235], and *Schizochytrium* sp. for pacific white shrimp [236], common carp [237], tilapia [238], and Atlantic salmon [239]. The inclusion of n-3 PUFA-rich microalgae not only enriched the food products with n-3 PUFA and improved n-6 to n-3 PUFA ratio but also generated other positive effects, for example, increased survival and growth rates [240]. Microalgal species with high n-3 PUFA levels, such as Veramaris microalgae products, are suitable for feeding fatty fish like salmon, and the flesh fat (muscle lipids) of this fish species has a high content of triacylglycerols, with the fatty acid composition being closely resembles dietary fatty acid profile [9,241]. In contrast, microalgae with a low content of n-3 PUFA appears to be ideal for feeding the species with a low level of lipids in their flesh in which the dominating class of lipids is phospholipids rather than triacylglycerols [241]. Fish have long been adapted to the natural abundance of n-3 PUFA and their ability for synthesizing n-3 PUFA, especially EPA and DHA, is insufficient. The addition of n-3 PUFA, EPA, and DHA is nowadays a routine practice in aquaculture, and the level of dietary n-3 PUFA in aquafeed is species-specific or determined by the flesh lipid levels and the product applications.

The replacement of fish oil n-3 PUFA in fish diet is achieved primarily by substituting fish oil with microalgal biomass, while oils extracted from microalgae have also been tested. For example, oils extracted from *Schizochytrium* sp., *Crypthecodinium cohnii*, and *Phaeodactylum tricornutum* have been tested successfully on gilthead seabream diet to replace fish oil [242]. Thraustochytrid *Schizochytriu* sp. oil was found to be an effective substitute for fish oil in the diet of Atlantic salmon and increased the amount of DHA in the muscle [243]. The replacement of fish oil with a DHA-rich oil extracted from *Schizochytrium* sp. algal meal at 11% in the diet of Atlantic salmon post-molts resulted in similar DHA levels, but a low level of EPA in the muscle [239]. The cell-disrupted microalgal biomass was found to be more effective than non-disrupted microalgae in enriching muscle n-3 PUFA [244], which might be attributed to the improved digestion and absorption rates. As the fatty acid composition of fish meat highly resembles that in the diet, EPA and DHA should be provided sufficiently in the diet in order to produce fish meat with high levels of EPA and DHA.

Similar phenomena were observed in livestock feeding with n-3 PUFA-rich microalgae [11] (Table 3). The addition of *Aurantiochytrium* sp. in dairy cows increased n-3 PUFA in milk [245], *Isochrysis galbana* and *Nannochloropsis* sp. in laying hen diet increased n-3 PUFA content in eggs. The inclusion of *Schizochytrium* sp. in lamb [246], goat [247], pig [248], and boar [249] diets improved n-3 PUFA contents and the ratio of n-6 to n-3 PUFA in the meat products. Subsequently, consumers obtain n-3 PUFA by taking food products derived from fish and animals that are supplemented with n-3 PUFA-rich microalgae biomass or oil extracted from microalgae.

Presently, due to limited production and high cost of microalgal biomass or extracted oils, the application of microalgae as a source of n-3 PUFA in terrestrial animal diet is not currently feasible. It is promising to use microalgae or extracted oil from microalgae in fish diet to meet fish n-3 PUFA requirements and enrich muscle lipid with n-3 PUFA for human consumption. From the point of view of production cost, it is more feasible to include microalgal biomass rather than extracted oil as a means of replacing fish oil. In addition to enriching fish muscle with n-3 PUFA, microalgae-based feed has also been reported to help fish develop faster and produce better fillets [250]. The unpredictable supply and extremely high production costs make their global use currently a big challenge, but it is believed that this issue will be resolved with the advances in cultivation system/technology and selection of species and strains, along with the further development and application of molecular and bioengineering techniques.

## 5. Sustainability

The consistent rise in the global population is pressing for developing novel and alternative sources of foods and functional ingredients to meet the rising market demand for high quality edible oils, especially n-3 PUFA. It is well understood that terrestrial crops are not good sources of n-3 PUFA as the major n-3 PUFA in plant sources are ALA, LA, or ARA. The terrestrial animal products contain n-3 PUFA with better profiles, for example, containing EPA and DHA, but the contents are low or quite low. Naturally, marine species dominate the supply of EPA and DHA. Global marine fisheries catches have been stable or slightly decreasing in the past decades due to the maximal or over harvesting and the climate changes. Inversely, aquaculture has been increasing fast in the past two decades to fill up the shortage of food supply. Aquaculture species, particularly fish, also require a high amount of n-3 PUFA in their diets for health and productivity, while enriching the products with n-3 PUFA for human consumption. As such, aquaculture actually competes with humans and animals, especially pets, for n-3 PUFA or n-3 PUFA-rich oils, conventionally from fish. To overcome this challenge, microalgae have been picked up as a promising source, and this scenario is supported by the available information on the suitability and sustainability of microalgae as solutions of producing n-3 PUFA-rich oils and many other nutritional compounds for food and feed products [251,252,253,254].

### 5.1. Techno-Economic Sustainability

To date, most research on the techno-economic sustainability of microalgae production focuses on the production of microalgae biofuel oil [255,256,257,258]. However, there are some assessments on the economic sustainability of high value-added microalgae edible products and impacts of different final products, production systems, microalgae species, and production areas [259]. In terms of cultivation systems, traditional open pond has a low cost and conversion rate of CO_2_. Photobioreactors have a higher cost, while being more efficient and having a higher productivity. At present, there are a few commercial manufacturers in the world that produce high value products using microalgae. Heterotrophic fermentation system is widely used by industry because of its higher productivity and is, therefore, the main culture system for the production of algal n-3 PUFA-rich oil. Tabernero et al. (2012) assessed the economic sustainability of an industrial fermentation system and reported that the cost for the production of *Chlorella protothecoides* biomass was about 1.23 € kg^−1^ [260]. Hoffman et al. (2017) showed that an algal turf scrubber system yields biomass at $510 tonne^−1^, while an open raceway pond (ORP) system yields biomass of the same species at $673 tonne^−1^ [261]. A recent study assessed the economic potential of microalgae as food ingredients, based on a photobiorector system for the cultivation of *Nannochloropsis* sp. at the industrial scale in Central Germany over a time span of 30 years [152]. The cultivation in photobioreactors in a cold-weather climate showed that microalgae compared favorably to fish aquiculture. In comparison with Atlantic salmon (*Salmosalar*) raised in aquaculture, EPA from *Nannochloropsis* sp. resulted in about half the cultivation cost (44–60%). Major expenditures comprised photobioreactor infrastructure, maintenance, and labor cost. Higher net present value and return-on-investment are expected to increase substantially when the cultivation occurs in a region with warm climates and extended cultivation seasons [152].

Further analysis covering the whole production processes/steps from cultivation to the final oil product indicates that the cost of microalgae oil is strongly impacted by the energy-intensive processes, from harvesting to oil extraction and, further, to oil purification. A quantitative analysis of algae harvesting–dewatering concluded that 3–30% of the total production costs come from this step [262,263], while the major costs occur in the downstream processing processes. Cell rupture turns out to be a critical issue for most of the oleaginous species in biomass processing [264]. In addition, avoiding expensive algae paste drying and development of lipid extraction without organic solvents could improve the processing efficiency and reduce the cost and environmental impact. The cost for bio-refining of the common agricultural biomass is between 20% and 40% of the total production costs, but for microalgae due to underdeveloped technologies, this cost increases up to 50–60% [262].

The cost for the production of microalgae oil is still high compared with vegetable oil and fish oil. However, the consumers of high-value food supplements (such as n-3 PUFA capsules) are willing to pay more for the same product and prefer products obtained from sources that are suitable for their particular eating habits (i.e., veganism, religious requirements), greater environmental awareness, and perhaps to a lesser extent, the allergic issues. Microalgae-derived n-3 PUFA-rich oil supplements meet these particular consumer requirements, and this advantage has been taken into consideration in the investment in the large scale production and exploiting whole microalgae as food ingredients. Heterotrophic cultivation is a promising mode of operation for cost reduction that involves a lower requirement of land and investment due to the smaller ratio of surface area to volume, wide range of carbon and energy sources, and reduced downstream processing costs [265]. It is apparent that there still are several challenges that affect techno-economic aspects of algal n-3 PUFA production, mainly the cultivation systems, productivity, and processing equipment and technologies. When the overall cost of production system is reduced and the techno-economic performance is moved up to a level that is close to that of fish oil, a broader adoption of algal n-3 PUFA production and commercialization will be achieved. In this regard, bioengineering potentially plays a very important role by increasing cell growth and density and elevating the cellular content of n-3 PUFA, EPA, and/or DHA.

### 5.2. Socio-Economic Sustainability

The algae industry is not older than 70 years and significant advancements have been made in the past decade. This industry mainly produces extracts for processed foods and other industries, such as cosmetics and medicine, while the ready-to-use of raw biomass has been widely generated as well. Similar to other industries, a sustainable algal industry can create a great value chain, which is an important step towards blue-green bioeconomy [266,267]. The socioeconomic impact of microalgae industry can be assessed and predicted from numerous aspects, including but not limited to societal factors, such as education, technical training, new businesses, startup companies and supports, financial investments, new sources of functional ingredients, and the improvement of life quality, and thus reduced health care costs associated with the consumption of microalgal products. It also includes research investment and activities, algal strain collection and characterization, technology and equipment developments, and intellectual property.

It is well known that disease not only affects life quality and lifespan but also leads to substantial economic loss through different ways, particularly productivity, treatment cost, and even mortality. The impact of clinical and subclinical diseases on labor availability and quality, production efficiency and economic returns may be greater than the losses as a result of mortality. This specific socioeconomic impact could be ignored to certain extent until the severe conditions occur, for instance, when chronic conditions progress to a level that results in illness, disability, and death. Nevertheless, the negative socioeconomic impact actually happens over a long time course until being recognized. To reduce the socioeconomic impact of chronic health conditions and diseases, the availability and affordability of nutritional and functional products are crucial by means of offering disease prevention and treatment through diet habit or intake of healthy food and other forms of health products. Among a number of naturally-occurring health products, n-3 PUFA, especially EPA and/or DHA, have been intensively examined for their health benefits (see Section 3 for details) and increasingly consumed by people in a wide age range from infancy to old age. Accordingly, the production and consumption of n-3 PUFA promotes, in several aspects, the socioeconomy of a community, country, and the world [268].

The global n-3 PUFA market has been growing fast and is predicted to keep on growing in the future. The largest market segments by application are, respectively, functional foods, dietary supplements, infant formulae, and pharmaceuticals. In terms of market value, the largest segment falls to the concentrates because of their high prices. As such, the key suppliers have developed concentrates with EPA and DHA up to 90%, which are mainly used in the nutraceutical and pharmaceutical applications [269]. For decades, fish oil dominated the n-3 PUFA market in both volume and value. However, algal n-3 PUFA has been increasingly added to this market, although the share is currently still minor. The industry leaders produce high grade algal EPA and DHA using heterotrophic microorganisms that are grown on sugar in closed fermentation vessels. The trend of depleting fish or marine stocks and increasing concerns about the sustainability of marine sources offer an opportunity of microalgae as a promising alternative of fish n-3 PUFA. Moreover, the absence of fishy smell and taste and labels like “vegetarian/vegan”, “Kosher”, or “organic” make algal oil superior over fish oil and provide options for the vegan (sub)population to benefit from consuming n-3 PUFA [269]. These appealing properties of algal n-3 PUFA will drive the investment and innovations and thus on the techno-economy, which in turn improves the socioeconomy of algal n-3 PUFA industry in connection with various applications, market sectors, and the whole value chain.

Regarding the economic geography, many microalgal farming infrastructures are built and operated in Asia, North America, Europe, and Australia. Germany, France, and Spain host the largest number of microalgae producers in Europe, and France dominates the production landscape [270]. In China, the algal industry has a significant impact on promoting regional economic and employment growth. Because of differences in the regional industrial structure, product distribution, and marginal consumption propensity, the regional economic and employment impacts vary significantly [271]. Although not the same, the production of biodiesel and n-3 PUFA using microalgae shares great similarities, differing primarily in the final products that can be directed by selecting specific microalgal species for different lipid profiles and cultivation methods. Coastal zones, particularly the locations around the equator, are considered to be the optimal cultivation sites due to stable annual light, temperature, and ready availability of seawater. A recent study screened a subtropical microalgal collection and identified a Chlorella strain MEM25 with a robust growth in a wide range of salinities, temperatures, and light intensities [272]. This species was evaluated for its economic viability for the production of PUFA and essential amino acids as food additives and performance in different scale cultivation systems from close photobioreactors to open race ponds at coastal zones under geographically specific conditions. The evaluation highlighted an appreciable profitability of MEM25 production for human and animal food using the open raceway pond systems [272]. Although the evaluation was not specific for n-3 PUFA production, the available information indicates great geographical differences of the socioeconomic impact of the microalgae industry worldwide, in a region or in a country.

Investment or funding support for microalgal research and technology development is another important factor to the socioeconomic impact of algal industry, including n-3 PUFA production. The current European political priorities favor a transition towards a more sustainable economy where developing algal sector has become its green and bioeconomy strategies [266]. The European Union funds algal projects, for example, a €8 million project under the European Union’s flagship climate change program to help Kanembu communities in Chad to deal with the impacts of climate change and develop renewable energies, and part of the project is to develop technologies for more effective cultivation and drying of Spirulina (https://ec.europa.eu/international-partnerships/stories/harvest-hope-spirulina-lake-chad_en. accessed on 23 March 2022). As an ecofriendly, nutrient-rich dietary supplement for humans and animals, Spirulina has been promoted as a possible solution to food insecurity and malnutrition in developing countries. The increased investment occurs not only in the research funding support, but also in the infrastructure for the commercial production of algal products. Among many examples is the opening of a new plant in Nebraska of the United States by Veramaris, a joint venture of DSM and Evonik in 2019, costing $200 million, to produce n-3 PUFA oil using microalgae Schizochytrium sp. Pond Technologies, and its Stelco Lake Erie Works facility in Ontario, Canada received funds in 2019 from the Inventiv Captical Management for a total of $16 million to build/expand the algal production facility (www.agfundernews.com; accessed on 25 March 2022).

To date, few studies have been conducted to systemically evaluate the socioeconomic impact of algal n-3 PUFA industry, while a number of studies have been carried out to assess the socioeconomic impact of algal industry for biofuel [266,273]. Recently, a socioeconomic assessment was conducted for the PUFA Chain by researchers at Wageningen University and Research, and a report documented the details of approaches, methods, and the results of the assessment [274]. The study included a macroeconomic assessment, a Life Cycle Costing or microeconomic analysis, followed by an overall analysis of strengths, weaknesses, opportunities, and threats, with emphasis on the production costs, current market prices or economic returns, and challenges in Northern, Southern, and Central Europe based on a heterotrophic cultivation system. The study concluded that the selection of specific algae strains holds the promise of producing pure DHA and EPA, which enable the precise dosing for specific applications. Heterotrophic system is used to produce DHA, which is marketed as a high value product, such as infant formula, because of high cost. The possibility of producing pure DHA and EPA on an industrial scale out of phototrophic algae is still under investigation. Apart from quality, affordability is critical to the general public, in particular those who are in the lower socioeconomic class. There is still a long way to go before algal n-3 PUFA become available and acceptable to the majority of the world population.

There are an increasing number of studies supporting the use of microalgae for the production of n-3 PUFA and other value-added ingredients and biomolecules, which also bring various socioeconomic benefits [212,275]. Although there is a strong potential of establishing sustainable algal industry for n-3 PUFA production, marketing, and consumption, information on the socioeconomic impact is limited. Therefore, further research is warranted to systemically assess the socioeconomic impact of this industry globally and in different geographic regions or countries with different climates and technological advances and availability.

### 5.3. Environmental Sustainability

In the past few decades, wild fish have been the most common source of n-3 PUFA. However, with the increase of global population and consumption per capita linked to the increased awareness of health benefits, apparently fish oil cannot meet the market requirement. Fisheries catches are maximized or over-exercised, limiting further increase of fish oil production from marine sources. In the meantime, aquaculture has its own issue to solve in competing for sources of DHA and EPA as feed. The reality is that the global average intake of DHA and EPA is still lower than the recommended levels [276,277]. In order to meet the growing demand for n-3 PUFA, the food industry has been seeking alternative sources and gradually investing in the large-scale microalgae productions.

Microalgae have been considered sustainable sources of n-3 PUFA and other special fatty acids because of their ability to grow and accumulate high contents of fatty acids under adverse conditions [278]. They can even grow in non-arable and underutilized lands, freeing up fertile land for terrestrial crops [279]. Microalgae can use light as energy and CO_2_ in the atmosphere and various organic compounds as carbon sources for the photosynthesis of fatty acids and other biomolecules. It is apparent that microalgal cultivation can help to reduce CO_2_ in the air and alleviate the intensification of greenhouse impacts [280,281]. At present, about 1/3 of human food in the world is wasted, and the safe disposal of food waste is a great concern [282,283]. Waste treatment becomes one of the most challenging problems acknowledged in the United Nations Development Program (UNDP)-Sustainable Development Goals [222]. It has been reported that the environmental impact of commercial algal biorefinery (producing n-3 fatty acids, biofuel, and high protein products) is similar to that of traditional fish production [284]. In heterotrophic and mixed-nutrient microalgae culture, food wastes and food by-products (such as cheese whey permeate and molasses) are good sources of organic carbon and nutrients of microalgae [285,286]. When used to support algae growth, their impacts on the environment are reduced. A life cycle assessment on autotrophic and heterotrophic algal cultivation for the production of food and feed shows that high moisture algae extrusion obtained from heterotrophic cultivation result in the products that are more environmentally sustainable compared with animal foods, such as beef and pork [287]. The further analysis by the same research group on the different cultivation systems and microalgal species for oil production indicated the differential environmental impacts. With advances in the exploitation of more microalgal species, their structural characteristics, biochemical composition, cultivation, and processing technologies, the economic and environmental benefits, suitability, and sustainability of microalgae as biological platforms for the production of n-3 PUFA and other high value lipids and biomolecules will be increasingly realized.

## 6. Current Challenges and Future Perspectives

Today’s consumers are increasingly aware of the beneficial roles of n-3 PUFA in preventing, delaying, and intervening various diseases, such as coronary artery disease, hypertension, diabetes, inflammatory and autoimmune disorders, neurodegenerative diseases like Alzheimer’s and Parkinson’s diseases, depression, and many other ailments. The benefits of n-3 fatty acids on aging and cognitive functions are also one of the hot topics in scientific research and clinical studies. However, numerous longitudinal surveys in many countries revealed that a large portion of population worldwide does not have a sufficient intake of n-3 PUFA through diet. To resolve the insufficiency of n-3 PUFA intake, recommendations have been provided and promoted globally to increase the consumption of fatty fish and other seafood that contain high contents of n-3 PUFA, such as those from wild catches or aqua-farming, and also dietary supplementation of fish oil or n-3 PUFA-rich oils. With the increase of global population, the awareness of the health benefits associated with n-3 PUFA, and socioeconomic improvement worldwide, the market demand for n-3 PUFA-rich oils have been continuously increasing, especially EPA and DHA. The fisheries’ catches and aquaculture cannot meet such a fast-expanding market, and consequently a strong demand is rising in seeking alternative sources of n-3 PUFA-rich oils. In this regard, microalgae have been receiving increased academic and industrial interests due to their abilities in producing high yields of EPA- and/or DHA-rich oils and the advantages of high productivity, techno- and socio-economic sustainability, and environmental sustainability. *Crytthecodiniumcohnii*, *Schizochytrium* sp., and *Phaeodactylumtricornutum* are some of the microalgal species that have high productivity rates of n-3 PUFA-rich oil. Several n-3 PUFA delivery strategies have been developed, including food fortification, dietary supplementation, and the addition of n-3 PUFA-rich microalgae in food products. Feeding farmed animals with microalgae can enrich animal foods with n-3 PUFA. With the advances of scientific research and technological development, it is expected that new microalgal species will be discovered that are able to produce n-3 PUFA-rich oil with better productivities and economic values. When the cost of microalgal PUFA-rich oil is reduced and becomes comparable with fish oil or other sources of n-3 PUFA-rich oils, microalgal n-3 PUFA will be used increasingly by adults, in addition to infants.

## 7. Limitations

Although the questions that we were trying to answer in this paper were identified prior to the search of literatures as mentioned in Section 2, the retrieved studies were not screened strictly for eligibility, nor assessed for quality using the inclusion and exclusion criteria that could be defined before starting the literature search, collection, and analysis. As such, the extracted data from the selected studies were not synthesized systemically to generate a statistical summary of effect estimates using a meta-analysis method. Thus, the information and opinions presented in this paper may not be sufficiently evaluated and may potentially have biases. A systemic review and meta-analysis on the multi-aspects covered in this paper is imperative.

## Figures and Tables

**Figure 1 foods-11-01883-f001:**
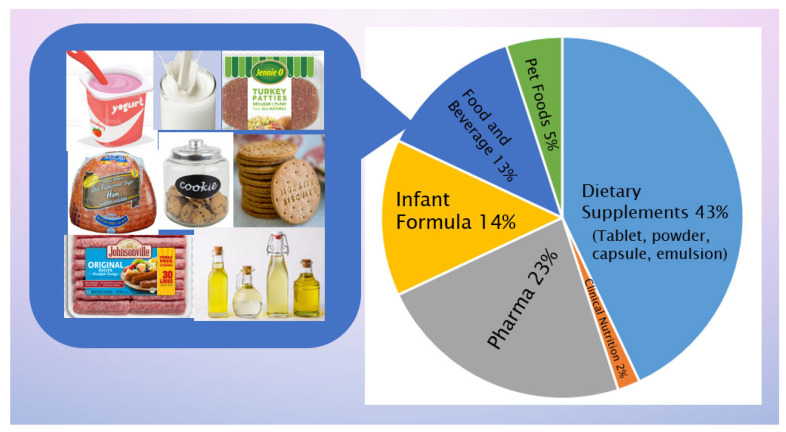
Application category and proportion of n-3 PUFA production (EPA/DHA) products in the global market (Modified from figure in [162]).

**Table 1 foods-11-01883-t001:** Summary of clinical trials for the effects of n-3 PUFA on neural development and neurodegenerative diseases.

Trial/Author Name	Daily Dose	Delivery Form	Duration	Outcome	Ref.
LipiDiDiet trial *	300 mg EPA/1200 mg DHA	Dairy drink	3 years	No effects after 2 years but improved after 3 years of treatment in 6 clinical tests related to cognitive function, brain atrophy, and disease progression in subjects with MCI.	[54,71]
Sinn 2012	EPA (1670 mg EPA + 160 mg DHA); DHA (1550 mg DHA + 400 mg EPA)	Capsule	6 months	Improved GDS scores in the EPA and DHA groups and verbal fluency in the DHA group in elderly people aged >65 years with MCI.	[48]
van de Rest 2008	High-dose (1093 mg EPA + 847 mg DHA); Low-dose (226 mg EPA + 176 mg DHA)	Capsule	6 months	No significant effect on cognitive performance in both the high-dose and low-dose groups.	[59,80]
Dangour 2010	200 mg EPA + 500 mg DHA	Capsule	24 months	Did not change the primary and secondary cognitive function outcomes.	[59,81]
Geleijnse 2011	240 mg EPA + 160 mg DHA	Margarine	40 months	Little or no cognitive decline observed during the study periods.	[59,82]
MAPT trial	800 mg EPA + 225 mg DHA	Capsule	3 years	No effects on cognitive decline in elderly people with memory complaints by multi-domain intervention and n-3 PUFA supplementation, either alone or in combination. Revealed by further analysis that participants with amyloid-β positive responded to the combined treatment or multi-domain intervention.	[72,74]
BENEFIC trial	30 g kMCT	Dairy drink	6 months	Increased brain ketone metabolism by 230% while it did not affect brain glucose uptake; improved episodic memory, language, executive function, and processing speed.	[75]
CARES trial ^#^	90 mg EPA + 430 mg DHA	Capsule	1 year	Increased tissue carotenoid concentrations and blood carotenoid and n-3 PUFA concentrations; tended to improve episodic memory and global cognition.	[79]
Yurko-Mauro 2010	900 mg DHA	Capsule	24 weeks	Doubled plasma DHA concentrations; improved PAL scores and immediate/delayed VRM scores, while it failed to improve working memory or executive function tests.	[83]
Bowman 2020	1650 mg EPA + DHA	Capsule	3 years	Failed to slow 3-year cerebral WMH accumulation and executive function decline in older non-demented adults with evidence of WMH.	[84]

* Souvenaid^®^ consists of other nutrients: uridine monophosphate, choline, vitamins (B12, B6, C, E, and folic acid), phospholipids, and selenium. ^#^ The intervention contained n-3 PUFA, xanthophyll carotenoids lutein, meso-zeaxanthin, zeaxanthin, and vitamin E. GDS, geriatric depression scale; kMCT, ketogenic medium chain triglycerides; MCI, mild cognitive impairment; PAL, paired associated learning; VRM, verbal recognition memory; WMH, white matter hyperintensities.

**Table 2 foods-11-01883-t002:** Microalgae species/strains used for n-3 PUFA production at large scales.

Species	EPA (%)	DHA (%)	Application	Company	Ref.
*Crythecodinium cohnii*		>50	IF *	DSM (NLD)/Martek BioSciences (US)	[153]
*Crythecodinium cohnii*		35	Suppl, IF	GCI Nutrient (US)	[154]
*Nannochloropsis Oculata*	25–26		Suppl, Food	Qualitas (ISR), Astaxa (DEU)	[154,155]
*Nannochloropsis Oculata*	>65		Pharm, Suppl	Aurora Algae (US)	[156]
*Phaeodactytum Tricomutum*	24	36	Suppl, Food	Nutraceuticals LLC DBA Valensa International (US)	[157,158]
*Schizochitrium* sp.	19	40	F&B	DSM (NLD), Veramaris (US)	[3]
*Schizochitrium* sp.	<2	43	IF	DSM (NLD)/OmegaTech (US)	[3]
*Thraustochytrium*		>40	IF, F&B	Mara Renewable (CAN)	[159]
*Ulkenia* sp.	11	44	Suppl, F&B	Lonza (CHE), Nutrinova (DEU)	[3,160,161]

* IF—infant formula, Suppl—supplement, Pharm—pharmaceutical, F&B—food and beverage.

**Table 3 foods-11-01883-t003:** Feeding farmed animals with n-3 PUFA-rich biomass or oil (Modified from table in [11]).

Microalgae	N-3 PUFA in Algae or Diet	Farm Animals	Outcome	Ref.
*Schizochytrium* sp.	31.3% Lipid; 14.3% n-3 PUFA, 5.3% DHA, and 0.3% EPA of TFA.	Atlantic salmon (*Salmosalar*, L.)	The replacement of fish oil with *Schizochytrium* sp. decreased the levels of persistent organic pollutants. When the algal biomass was included at 11% of the diet, the flesh fillet DHA levels reached similar levels to fish fed the fish oil diet.	[239]
*Cryptecodinum cohnii;* *Phaeodactylum tricornutum.*	1.37–1.59% DHA and 1.18–1.31% EPA in diet	Sparusaurata	Fish fed a diet containing 2% or 4% *Cryptecodinum cohnii* had a survival rate of 83.58%, significantly higher than fish fed a diet containing fish oil only (64.08%).	[240]
*Sparus aurata*, L.	3.48–7.02% DHA, 1.08–3.54% EPA, and 0.14–0.79%ARA in diet	Gilthead seabream	Fish fed diets containing DHA, DHA + ARA, or DHA + ARA + EPA showed higher n-3 PUFA contents. The weight and total length of larvae fed the diet containing DHA + ARA+ EPA in 2 weeks were higher than those fed the other diets.	[242]
*Thraustochytrid Schizochytrium* sp. L	13% Thraustochytrid oil diet; 60.0 % n-3 PUFA, 53.5% DHA, and 3.4% EPA of dietary TFA.	Salmosalar	DHA in muscle was increased from 3.0% to 6.0% in fish fed diets containing thraustochytrid oil compared with fish fed other diets.	[243]
*Isochrysis galbana* and *Nannochloropsis oculata*	120 mg microalgae per 100 g feed; 1.13–1.27% DHA of the dry algal biomass	Brown laying hens	The rigid cell wall of microalgae reduced the bioaccessibility of the n-3 PUFA in hens. DHA content in egg yolks increased from 40 mg/egg to 61–77 mg/egg after the supplementation of non-disrupted and disrupted algal biomass.	[244]
*Aurantiochytrium* sp.	0–0.6% microalgae of the diet; 12.3–16.4% n-3 PUFA and 0–3.06% DHA of TFA in diet	Dairy cow	Estimated intermediate doses (1.22 to 2.90 g/kg of DM) of DHA-rich microalgae (*Aurantiochytrium* sp.) could be beneficial to milk, fat-corrected milk, and energy-corrected milk yields, and is recommended for dairy cows.	[245]
*Schizochytrium* sp.	0–3% microalgae in diet; commensurable 0–0.75% DHA in diet.	Lamb	Carcass characteristics were unaffected; Daily DMI increased by the supplementation of PUFA-rich microalgae; The content of total PUFA in adipose tissue increased from 1.2 to 2.0 g/100 g fresh tissue; The ratio of n-6/n-3 PUFA in adipose tissue decreased from 219 to 44.	[246]
*Schizochytrium* sp.	4.16–8.44 g DHA and 1.65–3.34 g EPA/day in diet.	Goat	DHA and EPA contents were increased from 0.52% to 9.98% and 0.44% to 2.89%, respectively, in plasma lipids; The milk contents of C14:0, C16:0, trans-10 C18:1, trans- 11 C18:1, cis-9, trans-11 C18:2, trans-10, cis-12 C18:2, DHA, and DPA were also increased in goats supplemented with the algal biomass.	[247]
*Schizochytrium* sp.	7.03 g DHA, 3.13 g DPA, and 19.27 g PUFA g/kg of feed	Pig	N-3 PUFA was increased from 1.49% to 5.53% in longissimus thoracis et lumborum muscle fat, from 1.59% to 5.46% in semi-membranous muscle fat, and from 1.07–1.55% to 6.24–7.54% PUFA in adipose tissues.	[248]

## Data Availability

Not applicable.

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
