# Peer review of "Health Benefits, Food Applications, and Sustainability of Microalgae-Derived N-3 PUFA"

_foods, 2022, doi:10.3390/foods11131883_

Round 1
Reviewer 1 Report
The authors prepared an overview about health benefits, application and sustainability of microalgae-derived n-3 PUFAs. It is an interesting, hot topic, citing many exciting, recent researches but some tables/figures would help readers to see the main results. It would be useful to include a table about studies that have investigated the health effects of n-3 PUFAs (with type and dose of supplementation also) in neurodegenerative disease as well as in neurodevelopment and their results.
On page 3 (row 124-125) you cite the articles ‘in the 2018 trial’ and then you write ‘After a follow up of 7.4 years’. For me it is a little disturbing because these studies were published in 2018, but were performed earlier (VITAL: 2011-2017, ASCEND: 2005-2016). Maybe you should cite it as ‘the VITAL study published in 2018’. In row 128-9 it seems from your sentence that eating less fish can result in a 40% reduction of cardiovascular events. So eating less fish is more beneficial? In the original article this reduction was seen in the n-3 PUFA supplemented group. Please add this fact to the sentence also.
On page 8 (row 369-72) you write that microalgae-derived n-6 DPA is poorly studied to date, but Ref. 128 is only about n-3 DPA and Ref 129 about both n-3 and n-6 DPA. Please clarify it which DPA do you mean here!
The diverse food applications of microalgae-derived n-3 PUFAs could be summed up on a Figure to make it more visible for readers.
Some minor comments:
· - There are a lot of typos in the article (mainly in part 4; page 13-8), please correct them all in the whole manuscript.
· - Ref 30: in this study there are eight co-authors, not only two and the end of the title is also missing. Please correct it.
· - Ref 35: the 4th author is William Christen and not Christ Te N W. Please correct it.
· - Sometimes you write not only last names but also first names of authors (e.g.: Ref 36).
· - Ref 82 is updated in Ref 81, so please only cite the updated version (Ref.81) and delete the older one (Ref 82)
· - Please check the completeness and correctness of titles and authors in the whole references section.
Author Response
The authors appreciate the reviewer for taking time to go through our manuscript and providing constructive comments for improvement. Please see the attachment.

Reviewer 2 Report
Dear Authors,
I appreciated revising the paper “Health Benefits, Food Applications and Sustainability of Microalgae-Derived n-3 PUFA” you submitted to Foods.
The paper reports and in-depth insight into several aspects of n-3 PUFA including their effect on human health and the sustainability of their extraction from microalgae.
The main critical point of the paper is the lack for a systematic literature search. Hence, I suggest including a section reporting limitation of the paper. Moreover, in the Introduction section, the background of the paper and the novelty should be reported. Which knowledge does the paper add?
Please, mind that the abstract exceeds the allowed maximum word and Table 1 need formatting according Journal guidelines.
Final recommendation: minor revision.
Author Response

(The authors gave the same response as above.)

Reviewer 3 Report
Regarding the review entitled '' Health Benefits, Food Applications and Sustainability of Microalgae-Derived n-3 PUFA.'' This review is very interesting, well-written and organized. I have one comment.
Please add a table summarize the application of n-3 PUFA-rich microalgae in farm animals.
Author Response

(The authors gave the same response as above.)

Round 2
Reviewer 1 Report
The authors completely rewrote the manuscript according to the suggestions and it has improved a lot. The Figures and Tables makes the manuscript much clearer and easier to follow.
There are still typos in the article (e.g. rows 141, 145, 147: supplementation; row 149: individual; row 613: beverages; row 644,692 biomass; row 710: PUFA), please correct all of them.